# CaseEncoder: A Knowledge-enhanced Pre-trained Model for Legal Case Encoding

**Yixiao Ma**[*]
Huawei Cloud BU
Shenzhen, Guangdong, China
ma-yx16@tsinghua.org.cn

**Yueyue Wu**
Quan Cheng Laboratory
Institute for Internet Judiciary, Tsinghua University
DCST, Tsinghua University
wuyueyue1600@gmail.com

**Weihang Su**
Quan Cheng Laboratory
DCST, Tsinghua University
Institute for Internet Judiciary, Tsinghua University
swh22@mails.tsinghua.edu.cn

**Qingyao Ai**
Quan Cheng Laboratory
Institute for Internet Judiciary, Tsinghua University
DCST, Tsinghua University
aiqingyao@gmail.com

**Yiqun Liu** [†]
Quan Cheng Laboratory
DCST, Tsinghua University
Institute for Internet Judiciary, Tsinghua University
yiqunliu@tsinghua.edu.cn

## Abstract

Legal case retrieval is a critical process for modern legal information systems. While recent studies have utilized pre-trained language models (PLMs) based on the general domain self-supervised pre-training paradigm to build models for legal case retrieval, there are limitations in using general domain PLMs as backbones. Specifically, these models may not fully capture the underlying legal features in legal case documents. To address this issue, we propose CaseEncoder, a legal document encoder that leverages fine-grained legal knowledge in both the data sampling and pre-training phases. In the data sampling phase, we enhance the quality of the training data by utilizing fine-grained law article information to guide the selection of positive and negative examples. In the pre-training phase, we design legal-specific pre-training tasks that align with the judging criteria of relevant legal cases. Based on these tasks, we introduce an innovative loss function called *Biased Circle Loss* to enhance the model's ability to recognize case relevance in fine grains. Experimental results on multiple benchmarks

demonstrate that CaseEncoder significantly outperforms both existing general pre-training models and legal-specific pre-training models in zero-shot legal case retrieval. The source code of CaseEncoder can be found at `https://github.com/myx666/CaseEncoder`.

## 1 Introduction

Legal case retrieval is a critical process for modern legal information systems, as it aims to find relevant prior cases (i.e., precedents) that serve as important references to the case to be judged. In recent years, pre-trained language modeling (PLM) techniques have achieved great success in general-domain retrieval tasks, which has also led to attempts to apply PLMs in the legal domain. For example, Zhong et al. (2019) and Xiao et al. (2021) propose a legal-oriented PLM based on BERT (Devlin et al., 2018) and Longformer (Beltagy et al., 2020), respectively. However, existing legal-oriented PLMs have limitations in their adaptation to the legal domain, because they mainly rely on replacing general-domain training data with legal data or extending the input length to fit the long-length characteristic of legal documents. Ge et al. (2021) parse law articles in the form of premise-conclusion pairs to train a multi-level matching network for legal case

---

[*]This work is done when studying at Quan Cheng Laboratory & Institute for Internet Judiciary, DCST, Tsinghua University

[†]Corresponding Author

matching. Bhattacharya et al. (2022) propose Hier-SPCNet which substantially improves the network-based similarity by introducing legal textual information. However, they still adopt general-domain PLMs as the foundation of their approach. While interpretable, such models do not fully enable the PLM to comprehend legal concepts in case documents, which limits their generalizability and applicability.

To address this issue, this paper proposes CaseEncoder, a PLM that leverages fine-grained legal knowledge to improve both the data sampling phase and the pre-training phase. In the data sampling phase, we split law articles into unambiguous branches and construct inner logical relations in each article, which are used to estimate similarity weights between cases. These similarity weights serve as pseudo-labels to guide the selection of positive and negative cases. In this way, our proposed data sampling method improves the quality of sampled cases for further pre-training. In the pre-training phase, CaseEncoder adopts masked language modeling (MLM) and fine-grained contrastive learning tasks. These tasks aim to match two main concepts in the judging criteria of legal relevance: *key circumstances* and *key elements* (Ma et al., 2021b), respectively. *Key circumstances* refer to significant case descriptions in the document, while *key elements* are the legal-level abstraction of *key circumstances*, which focus more on consistency with law articles. In practice, the MLM task captures the semantic-level case descriptions, while the fine-grained contrastive learning task adopts sampled cases together with their relevance weights to enhance the model's ability to identify *key elements*. Based on the design of the fine-grained contrastive learning task, we propose an innovative loss function, Biased Circle Loss, which leverages the obtained fine-grained relevance score to optimize the recognition of *key elements*.

There are two advantages of annotating law articles as fine-grained legal knowledge. 1) **Cost efficiency:** law articles are usually close sets with a limited number of documents regardless of the legal system, and the number of law articles is much smaller than the number of legal cases in reality. Therefore, annotating law articles requires much less effort than annotating legal cases. 2) **Generaliability:** legal knowledge from articles is applicable to all cases under the corresponding legal system. Notably, the idea of annotating law articles

as fine-grained legal knowledge can be applied to all statutory law systems, or even to the common law systems, because statutory law is one of the most important sources of common law. Therefore, the effectiveness of the proposed methodology in this paper is not limited to a specific legal system.

Experimental results on multiple legal case retrieval datasets demonstrate that CaseEncoder significantly outperforms the existing general pre-training models and legal-specific pre-training models. We also present case document embedding visualizations to showcase the potential of CaseEncoder in downstream tasks such as charge prediction and article prediction.

## 2   Related Work

The methods for legal case retrieval can be divided into two categories: traditional bag-of-words methods and neural-based methods. Before the emergence of deep neural networks, TF-IDF+VSM (Salton and Buckley, 1988), BM25 (Robertson et al., 1995), and LMIR (Ponte and Croft, 1998) are representative bag-of-words methods. These methods have one common feature: they all treat documents as a sequence of terms and compute the document similarity score by statistical factors like term frequency or inverse document frequency. One limitation of such methods is that they ignore the information of the order among terms.

With the rapid development of NLP techniques, applying pre-trained language models (PLMs) to legal case retrieval has received huge success. Westermann et al. (2020); Shao et al. (2020) directly retrieve legal cases through a fine-tuned BERT (Devlin et al., 2018). Zhong et al. (2019) and Xiao et al. (2021) use large Chinese legal corpus to pre-train BERT and Longformer, respectively. In addition to applying original PLMs to the legal domain, there are also works trying to incorporate legal knowledge into their proposed models. For example, Sun et al. (2022) propose Law-Match, a model-agnostic causal learning method that utilizes the knowledge in law articles to tackle legal case matching. Barale et al. (2023) integrate the retrieval, processing, and information extraction into a pipeline for the recognition of key information in refugee cases. Their model retrieves a total of 19 categories of items from refugee cases before conducting information extraction by the state-of-the-art neural named-entity recognition method. Bhattacharya

```
┌─────────────────────────────────────────────┐
│        Title: XXX's Case of Hazardous Driving │
│   Court: People's Court of XXX Zone, Shandong Province │
│ Case Number: Criminal Judgment (2018) Lu (XXXX) Criminal No. XXX │
│ Parties' Information                          │
│    Public Prosecutor: Shandong Province XXX People's Procuratorate │
│    Defendant Background: Defendant XXX, male, was born in XXX… │
│ Fact:                                         │
│    (Accusation of the Prosecutor) On December 18, 2017, the defendant XXX was │
│    drunk and drove an ordinary two-wheeled motorcycle along Fushun Road in Weihai │
│    City when it collided with a small car…    │
│    (Evidence) …and provided the following: 1. The arrest issued by the Public │
│    Security Bureau of XXX; 2. Testimony of witness XXX … │
│    (Facts Confirmed by the Court) After investigation, it was found that on │
│    December 18, 2017, the defendant XXX was drunk and drove … │
│ Holding:                                      │
│    The court believes that the defendant XXX violated the transportation management │
│    regulations and drove a motor vehicle on the road drunk, and his behavior │
│    constituted the crime of hazardous driving… │
│ Decision:                                     │
│    The defendant XXX is guilty of hazardous driving, sentenced to one month and │
│    fifteen days of detention…                 │
│                                    Judge XXX  │
│                                 March 30, 2018│
│                                     Clerk XXX │
└─────────────────────────────────────────────┘
```

Figure 1: An example of legal case document.

et al. (2022) incorporate domain knowledge-based document similarity into PCNet (Minocha et al., 2015) and propose a heterogeneous network called Hier-SPCNet for a better representation of document similarity. However, these neural-based works all require task-specific fine-tuning or alignment to reach relatively good performance. By comparison, our proposed method can be directly applied to different downstream tasks in the zero-shot setting.

## 3 Task Definition

Given a query case $q$, the task aims to retrieve relevant cases from a candidate list $L = \{c_1, c_2, ..., c_M\}$, where $M$ is the size of $L$, and rank them by the relevance to $q$. Each candidate case document in the list has three main components: 1. *Facts* are objective fact statements confirmed by the court based on the evidence provided by the defendant and plaintiff. These statements typically answer questions such as where, when, and how the case occurred. 2. *Holding* is the judge's opinion on the key arguments of the case. It explains the reasoning behind the judge's decision. 3. *Decision* contains the final judgment of the defendant including the charge, sentence, and articles involved. This component is the official outcome of a case. An example of a legal case document is shown in Figure 1.

In the real legal case retrieval scenario, $q$ is usually a case to be judged. Therefore, $q$ only contains the *Facts* of a case, while candidate cases are precedents with a complete document including a title, meta information, *Facts*, *Holding*, *Decision*, and related law articles. In this paper, we focus

primarily on retrieving cases under criminal law. To match the real legal scenario that the retrieval model might have never seen $q$ or its related cases before, in this paper, all models are evaluated in a zero-shot manner.

## 4 Method

This section outlines the design and implementation of CaseEncoder. Figure 2 illustrates the overall framework of the model. We begin by introducing the fine-grained case sampling method used for data preparation in Section 4.1. Then, in Section 4.2, we describe the pre-training tasks proposed in CaseEncoder.

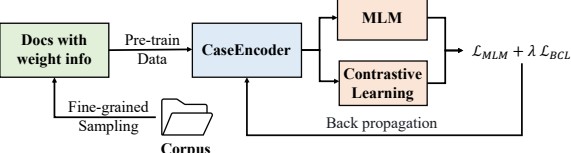

Figure 2: The overall framework of CaseEncoder.

### 4.1 Fine-grained Case Sampling

Recent studies on legal-oriented PLMs do not focus on understanding legal texts from a bottom-up approach. In other words, most PLMs simply replace the general-domain training corpus with legal texts without considering the legal correlation between these texts. This is mainly because the annotation of legal cases is time-consuming and requires much expertise, making it challenging to collect large-scale labeled data. On the other hand, in contrastive learning, which has proven to be effective in the pre-training phase, data needs to be sampled in advance as positive and negative cases. Unlike the general domain, in the legal domain, it is not appropriate enough to sample the positive and negative legal cases simply based on raw information in documents (e.g., charges, law articles, etc.). Because in a real legal scenario, a judge usually adjudicates a case based on how well *key circumstances* and *key elements* of the case match the constituent elements (fine-grained interpretation of the law article). Therefore, this paper proposes a fine-grained sampling method for legal case documents with reference to the specific process of judges deciding cases. By doing so, the sampled positive and negative cases can match the manually labeled relevance as much as possible for the subsequent contrastive learning task.

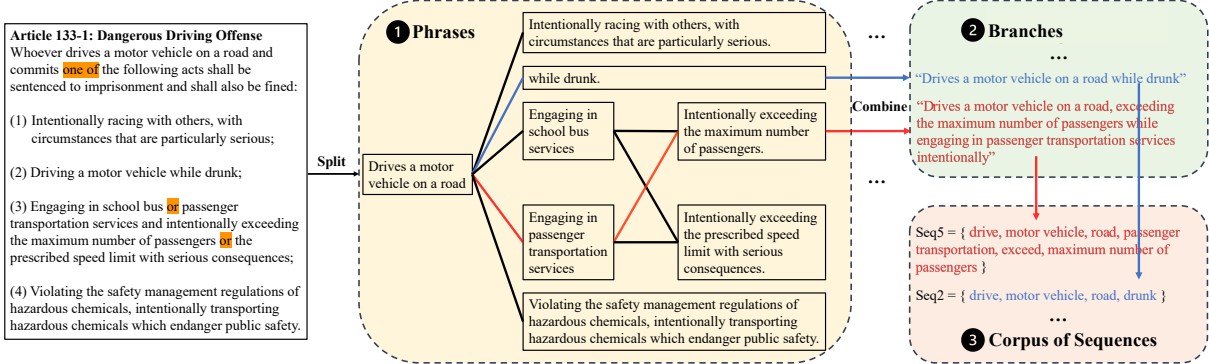

Figure 3: An illustration of the process of collecting a fine-grained sequence corpus.

Generally, a law article covers multiple branches. A branch describes a specific and unambiguous situation that conforms to the article. For example, *Article 133-1* from the Chinese Criminal Law shown in Figure 3 has four acts, among which act (1), (2) and (4) describes one certain branch each, but act (3) uses two 'or' clauses to cover all branches. Specifically, 'engaging in school bus service' and 'engaging in passenger transportation service' belong to different branches and will never appear in the same case even if they belong to the same article. In this paper, we call such phrases in a parallel relationship. According to the multiplication principle, act (3) describes $2 \times 2 = 4$ branches. Therefore, *Article 133-1* has $1 \times 3 + 4 = 7$ branches in total.

According to *Article 133-1*, phrases in one article can be written not only sequentially, but also in a parallel way. In other words, a law article is complexly structured. Even for two cases belonging to the same article, it is hard for models to recognize their legal relevance without any detailed information because such cases may belong to different branches of this article. To tackle this challenge, article information in legal cases requires further exploration in fine grains. In this paper, we identify more fine-grained article information, namely branch-level similarity, for each case as the preliminary of finding positive and negative cases in our case sampling algorithm.

To get branch-level similarity of legal cases, we first need to extract branches from all law articles. As described in Figure 3, the extraction and pre-processing include the following steps: First, all law articles are split into phrases. Then, the phrases belonging to the same article are reconstructed into branches. Similar to the example of *Article 133-1*, phrases written sequentially belong to the

same branch, while phrases in a parallel relationship belong to different branches. Finally, we split each branch into words and remove all meaningless words (e.g., stopwords, prepositions). The rest of the words in one branch are called a sequence. All such sequences belonging to the same law article constitute an article corpus. In conclusion, a law system with $N$ articles has $N$ article corpus:

$$C = \{C_{article\#1}, C_{article\#2}, ..., C_{article\#N}\} \quad (1)$$
$$C_{article\#p} = \{seq_{p1}, seq_{p2}, ..., seq_{pT}\}, p \in [1, N] \quad (2)$$

where $T$ is the number of sequences (i.e., the number of branches) in $article\#p$.

With the help of article corpus, we can obtain fine-grained legal features of cases. Specifically, we first extract *Holding* from the case document, which contains reasons supporting the final judgment and is highly related to the articles of the case. In the legal domain, cases committing the same crime have articles in common. Therefore, our case sampling strategy can recognize their fine-grained legal similarity with the help of article corpus. Suppose case $c_i$ belongs to $article\#p$ and the extracted *Holding* of case $c_i$ is denoted as $h_i$, we can compute the fine-grained legal feature $v_{ip}$ by:

$$v_{ip} = [f(seq_{p1}, h_i), f(seq_{p2}, h), \\ ..., f(seq_{pT}, h)] \in \mathbb{R}^T \quad (3)$$

where $f(seq_{pq}, h_i)$ denotes the BM25 (Robertson et al., 1995) score between the q-th sequence of $article\#p$ and $h_i$ given corpus $C_a$. The reason for using BM25 here is that experimental results show little difference between traditional and carefully designed complex methods in representing the term-level similarity. $v_{ip}$ can be intuitively interpreted as one feature of a legal case at the fine-grained article level. Next, given two cases $c_i$ and

$c_j$, the fine-grained similarity weight between two cases $w_{ij}$ can be presented as:

$$w_{ij} = \frac{|A_i \cap A_j|}{|A_i|} \times \max(\cos(v_{ik}, v_{jk}),$$
$$article\#k \in A_i \cap A_j) \quad (4)$$

where $A_i$ is the set of articles involved in case $c_i$, cos is the cosine similarity score, and $k$ represents the k-th article in $A_i$. In other words, the fine-grained similarity weight between two cases is mainly determined by two aspects: the extent of article overlap between two cases and the cosine similarity between their fine-grained legal feature $v_ik$ and $v_jk$. Note that a maximum function is added to Equation 4, as $w_{ij}$ is valued by the most similar situation of all articles in $A_i \cap A_j$.

Finally, for each case, $c_i$ in the legal case corpus, the fine-grained sampling method can sample a legally explainable positive case $c_{i+}$ according to the value of $w_{ii+}$, which can be regarded as a pseudo label indicating legal relevance. Since the fine-grained legal feature may be helpful to pre-training, the input data for the pre-training phase will be in the form of quadruples:

$$(c_i, c_{i+}, v_{ip}, v_{i+p})$$

where $p$ denotes the index of the law article with the maximum $\cos(c_{ik}, c_{jk})$ value in Equation 4.

## 4.2 Legal-specific Pre-training Task

As a law-oriented pre-trained model, we aim to integrate legal knowledge into the design of pre-training tasks, so that the model can acquire the ability to understand case documents not only at the semantic level but also at the legal concept level after training. To this end, in this paper, we refer to the judging criteria of relevant cases to design our pre-training tasks. As demonstrated by Ma et al. (2021b), two cases are relevant if they satisfy two requirements: high similarity between their *key circumstances*, and high similarity between their *key elements*. Specifically, *key circumstances* refer to important case descriptions, while *key elements* focus more on the consistency with law articles and represent the legal-level abstraction of *key circumstances*. In summary, a case is considered relevant to another when the case description and abstracted legal concept are both relevant.

Considering the overall performance in other NLP tasks, we adopt Chinese-RoBERTa (Liu et al., 2019) as the backbone of CaseEncoder. Following

the idea of the judging criteria, CaseEncoder is pre-trained by two tasks for the recognition of *key circumstances* and *key elements*, respectively.

### 4.2.1 Pre-training task 1

The first pre-training task is the masked language modeling (MLM) task, which enables the capture of regular semantic-level meaning of case description. As discussed in Devlin et al. (2018); Liu et al. (2019); Ma et al. (2021a), MLM contributes to producing embeddings with contextual information. Such embeddings are beneficial to the representation of *key circumstances* in legal cases. In detail, we only select *Facts* in a case document to randomly mask 15% of the tokens for MLM, because *key circumstances* are all included in *Facts*. The masked text is then fed into CaseEncoder to predict the masked tokens based on the surrounding unmasked tokens. The MLM loss function is defined as:

$$\mathcal{L}_{MLM} = - \sum_{x' \in m(\mathbf{x})} \log p\left(x' \mid \mathbf{x}_{\backslash m(\mathbf{x})}\right) \quad (5)$$

where $\mathbf{x}$ is the text in *Facts*, $m(\mathbf{x})$ is the set of masked tokens, and $\mathbf{x}_{\backslash m(\mathbf{x})}$ is the set of unmasked tokens.

### 4.2.2 Pre-training task 2

The second pre-training task is a fine-grained contrastive learning task that utilizes the information from quadruples $(c_i, c_{i+}, v_i, v_{i+})$ obtained in Section 4.1. The contrastive learning task in previous work (Chen et al., 2020) trains a model using augmented positive cases and regards the rest of the cases in the same batch as negatives. However, in the legal domain, the relevance scale is more fine-grained. One legal case can be partially relevant to another, and the extent of relevance is mostly determined by the previously mentioned *key elements*. Therefore, a fine-grained contrastive learning task is proposed to enhance the recognition of *key elements*. Specifically, suppose the batch size is $N$ and each quadruple has two cases, the total number of cases in a batch is $2N$. First, a multi-layer Transformer is adopted to obtain the representations of $2N$ cases. Then, we take the output of [CLS] token in the last hidden layer of Transformer as the case embedding: $e_1, e_2, ..., e_{2N}, e_i \in \mathbb{R}^H$, where $H$ is the hidden size. Finally, the training objective of this fine-grained contrastive learning task, Biased

Circle Loss (BCL), is defined as:

$$\mathcal{L}_{\text{BCL}} = \log[1 + \sum_{j=1}^{L} \exp(\gamma \alpha_n^j (s_n^j - \Delta_n))$$

$$* \sum_{i=1}^{K} \exp(-\gamma \alpha_p^i (s_p^i - \Delta_p))] \quad (6)$$

$$\alpha_p^i = |\exp^{w_p - 1} \cdot O_p - s_p^i|, \alpha_n^j = [s_n^j - O_n]_+ \quad (7)$$

where $s_p$ and $s_n$ are cosine similarity scores between positive and negative case embedding pairs, respectively. $\alpha_p$ and $\alpha_n$ are parameters controlling the speed of convergence, where $\alpha_p$ is determined by the legal-specific relevance weight in Equation 4. $\gamma$, $O_p$, $O_n$, $\Delta_p$, and $\Delta_n$ are hyperparameters of scale factor, optimum for $s_p$, optimum for $s_n$, between-class margin, and within-class margin, respectively. In this way, CaseEncoder is trained to pull case embeddings in the same class closer and push case embeddings in different classes apart. The distance between case embeddings in the vector space depends on the value of $s_p$ and $s_n$.

There are two main differences between our proposed loss function and Circle Loss (Sun et al., 2020): First, we expand the original loss function from a binary setting to a multi-class setting since legal cases within a batch can be classified into multiple classes. In detail, we consider any two cases to be in the same class if their legal-specific relevance weight is larger than a particular threshold $W_T$, and such a rule is transitive across all cases in a batch. Therefore, the actual implementation of Equation 6 is more complicated because cases are of multiple classes. Second, we add a weight parameter $\alpha$ to $\mathcal{L}_{\text{BCL}}$ to account for the extent of relevance between cases. As a result, the optimization object will shift depending on the value of $w$. By taking the legal-specific relevance weight into consideration, CaseEncoder is trained to discriminate between relevant cases in fine grains.

Finally, CaseEncoder is optimized by the linear combination of MLM loss and BCL loss:

$$\mathcal{L}_{\text{total}} = \mathcal{L}_{\text{MLM}} + \lambda \mathcal{L}_{\text{BCL}} \quad (8)$$

where $\lambda$ is a hyper-parameter

# 5 Experiments

## 5.1 Datasets and Evaluation Metrics

This paper adopts three publicly available datasets: LeCaRD, CAIL2021-LCR, and CAIL2022-LCR.

LeCaRD (Ma et al., 2021b) is the first Chinese legal case retrieval benchmark, which is widely used for the evaluation of retrieval models. **Challenge of AI in Law (CAIL)** (Xiao et al., 2018) is a competition held annually to promote AI technology and a higher level of digital justice since 2018. CAIL2021-LCR [1] and CAIL2022-LCR [2] are two competition datasets of CAIL.

Notably, since the experiment setting is zero-shot as introduced in Section 4, no training or fine-tuning stage is required for all methods mentioned in this paper. Therefore, both the training set and test set in the datasets are included for the evaluation. The total number of law articles involved in these datasets is 537. As a retrieval task, all models in this paper are evaluated by the Normalize Discounted Cumulative Gain (NDCG) metric. In this paper, we consider three different NDCG metrics: NDCG@10, NDCG@20, and NDCG@30.

## 5.2 Baselines

For a comprehensive evaluation of CaseEncoder on legal case retrieval, in this paper, we adopt the following PLMs which are widely applied in the legal domain as baselines:

- **BERT-XS** (Zhong et al., 2019) is a new release of BERT (Devlin et al., 2018) which conducts a secondary pre-training on Chinese criminal documents. Compared to the original BERT, the BERT-XS mainly focus on legal domain-specific tasks.

- **Lawformer** (Xiao et al., 2021) is the first law-oriented and longformer-based (Beltagy et al., 2020) language model, which combines task-motivated global attention and local sliding window attention to capture long-distance information. Lawformer extends the input length from 512 tokens to 4096 tokens to adapt to the long length of legal documents.

- **BERT-PLI** (Shao et al., 2020) aggregates paragraph-level semantic similarity between two case documents to retrieve legal cases. As the backbone of BERT-PLI, a BERT model is fine-tuned by a case law dataset in COL-IEE2019 (Rabelo et al., 2019) for alignment.

[1] http://cail.cipsc.org.cn/task3.html?raceID=3&cail_tag=2022

[2] http://cail.cipsc.org.cn/task_summit.html?raceID=1&cail_tag=2021

Table 1: Evaluation results of different models on LeCaRD, CAIL2021-LCR, and CAIL2022-LCR. $\dagger\backslash\ddagger\backslash\S$ denote statistical significance compared to CaseEncoder at a level of p-value $< 0.05\backslash0.01\backslash0.005$ using Wilcoxon test.

| Model | LeCaRD NDCG@(10, 20, 30) | | | CAIL2021-LCR NDCG@(10, 20, 30) | | | CAIL2022-LCR NDCG@(10, 20, 30) | | |
|---|---|---|---|---|---|---|---|---|---|
| BERT-XS | $0.343^\S$ | $0.355^\S$ | $0.384^\S$ | $0.361^\S$ | $0.368^\S$ | $0.388^\S$ | $0.358^\S$ | $0.359^\S$ | $0.383^\S$ |
| BERT-PLI | $0.525^\S$ | $0.519^\S$ | $0.538^\S$ | $0.555^\S$ | $0.547^\S$ | $0.560^\S$ | $0.512^\S$ | $0.499^\S$ | $0.516^\S$ |
| Lawformer | $0.620^\S$ | $0.623^\S$ | $0.636^\S$ | $0.691^\S$ | $0.684^\S$ | $0.699^\S$ | $0.694^\S$ | $0.688^\S$ | $0.700^\S$ |
| RoBERTa | $0.748^\dagger$ | $0.762^\ddagger$ | $0.791^\S$ | $0.804^\dagger$ | $0.817^\ddagger$ | $0.850^\dagger$ | $0.793^\ddagger$ | $0.803^\S$ | $0.837^\dagger$ |
| RoBERTa-Legal | $0.742^\ddagger$ | $0.764^\S$ | $0.806^\S$ | $0.814^\dagger$ | $0.823^\dagger$ | 0.855 | $0.800^\ddagger$ | $0.811^\S$ | $0.846^\dagger$ |
| OpenAI API | $0.672^\S$ | $0.695^\S$ | $0.729^\S$ | $0.715^\S$ | $0.726^\S$ | $0.765^\S$ | $0.714^\S$ | $0.725^\S$ | $0.763^\S$ |
| **CaseEncoder** | **0.785** | **0.803** | **0.839** | **0.842** | **0.849** | **0.876** | **0.833** | **0.840** | **0.867** |

- **RoBERTa** (Liu et al., 2019) is a replication of BERT (Devlin et al., 2018) which extensively studies the setting of hyper-parameters and training data size. Experimental results on various NLP tasks show that RoBERTa can match or exceed the performance of many previous PLMs. Since all datasets in this paper are based on the Chinese law system, we adopt the Chinese version of RoBERTa (Cui et al., 2019) as the baseline.

- **RoBERTa-Legal** is a legal version of RoBERTa that conducts a secondary pre-training on legal data using the MLM task.

- **OpenAI API**[3] is an official and publicly available way to get text embeddings generated by OpenAI LLMs. In this paper, we adopt the second generation of embedding models provided by OpenAI, *text-embedding-ada-002* (Brown et al., 2020), to measure the similarity of legal documents.

## 5.3 Implementation Details

For a fair comparison, CaseEncoder and baselines are adopted to retrieve legal cases in the same manner. Specifically, all models adopt a commonly used dual-encoder paradigm to retrieve legal cases. That is, models take the full text of query and *Facts* part of candidates separately to generate document-level query embeddings and candidate case embeddings without any secondary pre-training or fine-tuning. The final retrieved ranking list is sorted by the cosine similarity between query embeddings and candidate case embeddings.

All PLMs are imported from Huggingface (Wolf et al., 2019), with the learning rate set to $1*10^{-5}$.

[3]https://platform.openai.com/docs/guides/embeddings

BM25 algorithm is implemented by Gensim (Řehůřek et al., 2011). The hyper-parameter for CaseEncoder is: $\gamma = 16$, $O_p = 1.25$, $O_n = 0.25$, $\delta_p = 0.75$, $\delta_n = 0.25$, $W_T = 0.25$, and $\lambda = \exp*10^{-6}$. All training and experiments are conducted on eight 32G NVIDIA V100 GPUs.

## 5.4 Experimental Results

The overall performance of CaseEncoder and baselines on LeCaRD, CAIL2021-LCR, and CAIL2022-LCR datasets are shown in Table 1. Through the experimental results, we have the following observations:

- Compared to baselines, CaseEncoder has the best results in terms of all evaluation metrics on three datasets. Besides, CaseEncoder outperforms baselines with statistical significance except for one result (i.e., NDCG@30 of RoBERTa-Legal on CAIL2021-LCR), but still by a large margin (i.e., 0.855 to 0.876). This phenomenon illustrates the effectiveness of CaseEncoder in retrieving legal cases without any fine-tuning. The knowledge-enhanced sampling strategy and pre-training tasks facilitate the retrieval of relevant cases with legal meaning.

- The overall results on LeCaRD are comprehensively lower than those on CAIL2021-LCR and CAIL2022-LCR datasets. This is mainly due to the heterogeneity of the data corpus. Specifically, the pooling of candidate documents in LeCaRD does not have any restrictions, while CAIL2021-LCR and CAIL2022-LCR filter out documents with extremely long length. Consequently, LeCaRD is more challenging than the other two datasets.

- As the backbone of CaseEncoder, RoBERTa has competitive results among all baselines, which is

Table 2: Ablation study on LeCaRD. 'w/o sampling' denotes CaseEncoder without the fine-grained sampling method. 'w/o BCL (infoNCE)' and 'w/o BCL (Circle)' denotes replacing BCL with infoNCE and Circle Loss, respectively. 'w/o task' denotes CaseEncoder only pre-trained by MLM task.

| Model | NDCG@(10, 20, 30) |
|---|---|
| CaseEncoder | **0.785**, **0.803**, **0.839** |
| w/o sampling | 0.778, 0.795, 0.827 |
| w/o BCL (infoNCE) | 0.741, 0.764, 0.805 |
| w/o BCL (Circle) | 0.762, 0.782, 0.816 |
| w/o task | 0.742, 0.764, 0.806 |

consistent with its performance on other NLP tasks. Furthermore, RoBERTa-Legal has the second-best overall performance and outperforms the original RoBERTa, which proves the idea in Gururangan et al. (2020) that a secondary pre-training using domain-specific data is beneficial to the overall performance in the target domain. Finally, CaseEncoder outperforms its backbone and the legal version of its backbone, which indicates that the effectiveness of CaseEncoder is not only due to the training data.

- The effectiveness of BERT-XS (Zhong et al., 2019) is limited because it utilizes Next Sentence Prediction (NSP) task for pre-training and its [CLS] token is not trained to represent document-level embeddings. Besides, BERT-PLI is not as effective as reported in Shao et al. (2020). One possible explanation is that BERT-PLI utilizes the dataset from the entailment task in COL-IEE2020 (Rabelo et al., 2020) as an external legal data source to fine-tune the model. For a fair comparison, no external data is imported for any further training in this paper. Therefore, the performance of BERT-PLI declines compared to its previous results reported in Shao et al. (2020).

- The embedding LLM provided by OpenAI is not as effective in zero-shot legal case retrieval as it is in the general domain. Its overall performance is even surpassed by PLMs with much fewer parameter scales such as RoBERTa and RoBERTa-Legal. This result indicates that currently, LLM in the general domain is not the best choice for legal case retrieval.

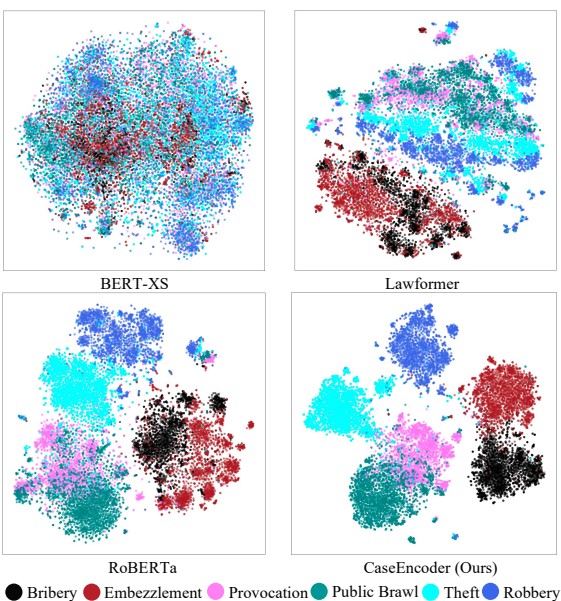

Figure 4: The visualization of case embeddings generated by four PLMs in the zero-shot manner.

## 5.5 Ablation study

To investigate the effectiveness of our proposed fine-grained sampling method, the knowledge-enhanced contrastive learning task, and our proposed BCL loss function, we further conduct an ablation study. In detail, we: 1) replace our sampling method with normal in-batch sampling (cases belonging to the same article are simply regarded as positive cases), 2) replace the BCL loss with infoNCE (Oord et al., 2018), 3) replace the BCL loss with Circle Loss (Sun et al., 2020) and 4) remove the fine-grained contrastive learning task, respectively. As shown in Table 2, replacing the sampling method, BCL loss function, or removing the contrastive learning task all lead to performance decline. Therefore, all of these innovations proposed in this paper contribute to the effectiveness of CaseEncoder. The result in Table 2 also indicates that BCL contributes most to the improvement of CaseEncoder. The traditional binary contrastive learning task has limited contribution to the performance because removing the whole contrastive learning task has a similar performance to replacing BCL with infoNCE, which means the improvement is mostly made by BCL.

## 5.6 Visualization of Case Embeddings

CaseEncoder is designed to effectively model case documents in the legal domain. In addition to the retrieval task, the document-level case embedding

can also be utilized in other downstream tasks such as charge prediction, sentence prediction, and law article recommendation. Therefore, the quality of document-level case embeddings is the basis of downstream task performances. Figure 4 is an example of how CaseEncoder improves the quality of case embeddings for charge prediction. The cases in Figure 4 belong to six criminal charges. For each charge, we randomly select 2,500 cases from the legal corpus and generate their case embeddings using four different methods. Then, we use t-SNE (Van der Maaten and Hinton, 2008) to reduce the dimension of case embeddings for visualization. Among all PLMs, CaseEncoder has the best ability to divide case embeddings into six clusters based on their charges, with only one pair of similar charges (Provocation and Public Brawl) having some overlap. By comparison, RoBERTa partially distinguishes between six charges, but with more overlap than CaseEncoder. The performance of BERT-XS and Lawformer is limited, which is consistent with the retrieval result and explanation in Section 5.4. These visualizations demonstrate how the fine-grained legal knowledge embedded in CaseEncoder can be leveraged for a range of legal applications beyond case retrieval.

## 6 Conclusion and Limitation

This paper proposes CaseEncoder, a pre-trained encoder that utilizes fine-grained legal knowledge to enhance the representation of case document embeddings. By introducing law article annotation into the sampling method, we improve the quality of sampled positive and negative cases during data preparation. Next, in the pre-training stage, CaseEncoder adopts two legal-specific pre-training tasks to align with the relevance judgment criteria in the legal domain. Experiments and visual analysis demonstrate the effectiveness of case embeddings generated by CaseEncoder in solving zero-shot legal case retrieval.

The main limitations of CaseEncoder are: Firstly, the definition of similar legal cases is based on the assumption that "cases committing the same crime have articles in common", which is not applicable to all legal systems. However, the idea of annotating law articles as fine-grained legal knowledge to enhance the performance of PLMs has the potential to be adopted in other law systems. In the future, we will consider validating the effectiveness of CaseEncoder on datasets of different law

systems and languages. We will also explore the potential of CaseEncoder for different downstream legal tasks. Secondly, the transitive rule in Section 4.2.2 is not consistent under all circumstances. For example, suppose case A and B are in the same class, and B and C are in the same class, then A and C still might have no law articles in common if case A and C both belong to multiple classes. According to our experimental results, the influence of this special case can be alleviated by reducing the batch size in the pre-training stage, because the possibility of two multiple-class cases appearing in the same training batch drops with the reduction of batch size. We will leave the optimization of our proposed BCL function as our future work.

## Acknowledgements

This work is supported by Quan Cheng Laboratory (Grant No. QCLZD202301) and the Natural Science Foundation of China (Grant No. 62002194).

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
