# OpenReview forum: "CaseEncoder: A Knowledge-enhanced Pre-trained Model for Legal Case Encoding"
_EMNLP/2023/Conference — EMNLP 2023 Main_

### Official Review · Reviewer_8osR · 2023-07-21

**Soundness:** 3

**Excitement:**

3: Ambivalent: It has merits (e.g., it reports state-of-the-art results, the idea is nice), but there are key weaknesses (e.g., it describes incremental work), and it can significantly benefit from another round of revision. However, I won't object to accepting it if my co-reviewers champion it.

**Paper Topic And Main Contributions:**

This paper studies the problem of legal case retrieval (finding relevant prior cases for a given case). It proposes domain-specific techniques to further pre-train RoBERTa with the MLM objective, as well as a contrastive objective tailor-made for legal cases. In particular, it modifies standard in-batch sampling with a sampling approach that uses domain expertise to measure the similarity between cases (e.g. how close two cases are consistent with the same law article). It then uses a modified version of Circle Loss (Sun et al., 2020) as the training objective, which is combined with the standard MLM objective.

The model is evaluated on three Chinese legal case retrieval datasets, and shows that it outperforms the RoBERTa baseline by 2 - 4 points. (Other baselines are also reported, but they often achieve much lower accuracy compared to the RoBERTa baseline.)

**Questions For The Authors:**

- What's the typical length of a sample in each of the evaluation dataset? Most of the models mentioned in this paper rely on BERT or RoBERTa encoders, which have max_seq_len of 512 or 1024 BPE tokens. Is that enough for legal cases?

**Reasons To Accept:**

- The proposed model consistently outperforms the RoBERTa baseline as well as other reported methods.

- The ablations are helpful in that it shows how much impact each ingredient has on the final performance (e.g. data sampling, training objective, etc.).

**Reasons To Reject:**

- This paper may not be suitable for the EMNLP venue for two reasons: 1) The proposed approach relies heavily on domain-specific techniques and the method is not general enough to provide much value for the wider NLP research community. 2) The reviewers may not have the necessary legal acumen to properly judge the claims of this paper. For instance, the authors justifies the generality of its main methodology by claiming that "the idea of annotating law articles as fine-grained legal knowledge can be applied ... even to the common law systems. Therefore, the effectiveness of the proposed methodology in this paper is not limited to a specific legal system.", something EMNLP reviewers may not be able to verify. This paper is probably better suited for a legal NLP conference or workshop.

- The paper focuses on a "retrieval" task but only considers embedding models among its baselines. It would be great to add comparison to standard IR models. For instance, the BM25 baseline, SoTA dense retrievers (e.g. DRAGON https://huggingface.co/facebook/dragon-plus-query-encoder), SoTA sparse retrievers (e.g. SPLADE++ https://github.com/naver/splade), or multi-vector retrievers (e.g. ColBERT-v2 https://github.com/stanford-futuredata/ColBERT). Complex domain-specific approaches are better justified if they're shown to significantly outperform off-the-shelf retrieval models.

- Some important details are missing. Please correct me in the rebuttal if I missed anything, but I failed to find any mention of the "pre-training data" used in this paper. It only mentions the evaluation datasets, but nothing is discussed on the data used to pre-train the model. What's the corpus for MLM? What's the case corpus used in the contrastive training?

**Reproducibility:**

2: Would be hard pressed to reproduce the results. The contribution depends on data that are simply not available outside the author's institution or consortium; not enough details are provided.

**Reviewer Confidence:**

2: Willing to defend my evaluation, but it is fairly likely that I missed some details, didn't understand some central points, or can't be sure about the novelty of the work.

---

> ### Author Rebuttal · Authors · 2023-08-28
>
> We appreciate your comments and suggestions, which we believe is beneficial to the refinement of our work.Due to space limit, we only address on the most important concerns:
> ### 1. Need more IR models as baselines.
> **A:** That is a good suggestion! However, due to the time limit of this reply period, we currently compare our proposed CaseEncoder to one of the best models, ColBERT-v2, as you metioned, on three datasets in the zero-shot setting to demonstrate the effectiveness of CaseEncoder:
> |                    | **LeCaRD**        | **CAIL21-LCR**     | **CAIL22-LCR**    |
> |:----------------------:|:-----------------:|:------------------:|:-----------------:|
> |                | NDCG@(10, 20, 30) | NDCG@(10, 20, 30)  | NDCG@(10, 20, 30) |
> | **ColBERT-v2**         | 0.715 0.743 0.803 | 0.814 0.805 0.851  | 0.791 0.780 0.837 |
> | **CaseEncoder (Ours)** | 0.785 0.803 0.839 | 0.842 0.849 0.876  | 0.833 0.840 0.867 |
>
> The result shows that CaseEncoder is comprehensively better than ColBERT-v2. Currently we are still doing experiments and the results of the rest baseline models will be reported in the camera-ready version of our paper.
>
> ### 2. Reproducibility
> **A:** The source code of CaseEncoder together with checkpoint are available on https://github.com/Anonymous-EMNLP2023/CaseEncoder. All datasets used in this paper are public benchmarks for legal case retrieval.
>
> ### 3. Rely on domain-specific techniques & not for NLP community
> **A:** Thank you for your question. We believe that legal domain is a very important part of NLP community. Currently various legal tasks (e.g., legal search, judgement prediction, legal text generation, and so on.) rely heavily on the NLP techniques.
>
> In addition, we emphasize that the techniques proposed in this paper is not restricted to the legal domain. The idea of sampling fine-grained pre-training data, BCL, and the fine-grained contrastive learning task are general NLP techniques adopted in the legal domain instead of pure legal research. Our proposed method can be applied to other NLP domains which require a fine-grained recognition of unlabeled documents in the pre-training stage.
>
> ### 4. Missing introcution of the pre-training data
> **A:** Thank you for your careful review! The pre-training data used in this paper are raw criminal judgment documents randomly sampled from online website, which is introduced in Section 3. However, we acknowledge that the current introduction is insufficient. We will add more details about the pre-training data in the camera-ready version.
>
> ### 5. Is the model input length enough for legal cases?
> **A:** According to our statistics, the average length of legal cases is over 2000. However, in this paper, only the **fact** part (<700 words) of case documents is input into models, which is widely accepted and adopted in previous works. This is because **fact** is the key part to describe the case and is enough for the pre-training of language models. The rest of a case document is used in other places, such as the reference for fine-grained sampling, the extraction of legal articles. Therefore, the model input length is enough for legal cases.

---

### Official Review · Reviewer_E28B · 2023-08-01

**Soundness:** 5

**Excitement:**

4: Strong: This paper deepens the understanding of some phenomenon or lowers the barriers to an existing research direction.

**Missing References:**


Another paper addressing past case retrieval, by means of targeted information extraction, that may be of interest to the authors: https://aclanthology.org/2023.findings-acl.187/

**Paper Topic And Main Contributions:**

This paper introduces CaseEncoder, an encoder designed specifically for legal documents that relies on introducing knowledge at the pre-training step and at the data sampling step. The targeted task is Legal Case Retrieval.

**Questions For The Authors:**

Do you have an explanation or a comment on the limitation on the similar case definition mentioned in the paragraph above? (I noticed there is no limitation section in the paper for now, it could be useful to add in the camera-ready and to answer this question there).

**Reasons To Accept:**

- PLMs may not be a good fit for legal documents as it fails to capture their specificities, because their aim is to have a broad coverage. This is the starting point of the paper and the identified problem it aims to solve. This is very relevant to the current domain-specific NLP literature and concerns. The authors seek to re-balance this by introducing a layer of legal knowledge, and aiming for a more domain-targeted pre-training. Literature has indeed shown that models that are closer to a domain, eg legal, despite their smaller size, may still perform better on a range of tasks.
- Very precise task definition in section 3
- Section 4.1: the fine-grain case sampling aims at addressing the usual bottleneck of explicit supervision, using "branch level similarity". This is very relevant to the legal domain in particular that suffers from lack of resources.
- The idea of introducing legal pre-training is interesting, while most legal NLP work have been focusing on fine-tuning PLMs.
- Evaluation is made on several benchmarks of legal NLP and CaseEncoder always performs better.

Overall, this is a very interesting paper, well written and clear, that explores a new way of introducing knowledge and of fitting the specific needs and challenges of legal NLP.

**Reasons To Reject:**

- Section 4.1: because the sampling is based on law article, with the aim of retrieving similar cases, the authors make the assumption that "cases committing the same crime have articles in common" line 277. One could argue that this may not always be the case, depending on the jurisdiction, the type of law if civil or common, and the area of law (e.g. international law). Another drawback is that it would not allow to differentiate among cases that include the same "crime". This rather limited and bounded definition of case similarity is a limitation.

**Reproducibility:**

4: Could mostly reproduce the results, but there may be some variation because of sample variance or minor variations in their interpretation of the protocol or method.

**Reviewer Confidence:**

5: Positive that my evaluation is correct. I read the paper very carefully and I am very familiar with related work.

---

> ### Author Rebuttal · Authors · 2023-08-28
>
> We do appreciate your recognition of our work! The suggestions are helpful to the refinement of our work. Due to space limit, we focus on addressing the most important concerns:
> ### 1. The assumption depends on the type of law if civil or common
> **A:** Thank you for your suggestion! We acknowledge the assumption has limitations. However, in addition to civil law, our proposed method can also be applied to common law if the judgment basis in the assumption is switched from article to legal issues. Specifically, we need to first prepare common law data with annotated issues to pretrain our CaseEncoder, with the rest model design remains unchanged. In this way, the idea of recognizing relavant legal cases in fine grains is adaptive to different legal systems.
>
> ### 2. To differentiate among cases that include the same "crime"
> **A:** Cases committed the same crime can still be differentiated because we focus not only on the overlap of legal articles, but also on the similarity of key judgment sentences in two case documents. In this way, relevant cases can be recognized in fine grains. The idea of sampling fine-grained cases is introduced in Section 4.1. We will add more explanations and details to this section in our camera-ready version.
>
> ### 3. Missing limitation and references
> **A:** Thank you for your careful review! We will discuss the limitation of our work(including your first question) and add missing references in our camera-ready version.

---

### Official Review · Reviewer_aaUB · 2023-08-04

**Typos Grammar Style And Presentation Improvements:** 1. Section 4.2 is too long. I suggest…
**Soundness:** 4

**Excitement:**

4: Strong: This paper deepens the understanding of some phenomenon or lowers the barriers to an existing research direction.

**Paper Topic And Main Contributions:**

This paper focuses on legal case retrieval, which aims to find relevant legal cases for a specific case. It utilizes law articles for pretraining, which enables PLM to learn legal concepts in legal cases. Specifically, to sample positive and negative cases, it utilizes different branches to construct branch-level similarity for the contrastive learning task. Together with the traditional MLM task, the PLM is able to capture key elements and key circumstances information in the cases. The experimental results show the effectiveness of the proposed model.

**Questions For The Authors:**

A) Line 406: since contrastive learning aims to distinguish positive pairs from negative ones, I wonder why the loss function should be in the multi-class setting.

B) Line 408 to 411: this transitive rule seems to lead to inconsistency. For example, A and B are in the same class, and B and C are in the same class, but these two facts do not necessarily imply that cases A and C are in the same class via your rule.

**Reasons To Accept:**

1. The idea of splitting legal articles into branches is interesting, and I think it is potentially valuable for future studies on this task.
2. The proposed branch-level similarity is a simple yet effective method to measure the relevance between cases and articles, which shows helpful to the contrastive learning process.
3. Extensive experiments demonstrate the effectiveness of the proposed method.

**Reasons To Reject:**

1. Although this paper mentioned two important kinds of concepts, i.e., key circumstances and key elements, they are indirectly modeled via two pretraining tasks. In addition, this paper does not provide any case study to show that the model indeed comprehends these two kinds of concepts better.
2. Some parts of the model section are a little confusing.

**Reproducibility:**

3: Could reproduce the results with some difficulty. The settings of parameters are underspecified or subjectively determined; the training/evaluation data are not widely available.

**Reviewer Confidence:**

3: Pretty sure, but there's a chance I missed something. Although I have a good feel for this area in general, I did not carefully check the paper's details, e.g., the math, experimental design, or novelty.

---

> ### Author Rebuttal · Authors · 2023-08-28
>
> Frist of all, thank you for your review! The review comments will substantially improve our work. Due to space limit, we focus on addressing the most important concerns:
> ### 1. Lack of case study to show the uesfulness of key circumstances and key elements
> **A:** We have now added a case study to prove the usefulness. Due to space limit, we only show the main results:
> ```
> Given:
> - Query case id=3746
> - Doc id=23502, relevance label = 0 (negative)
>
> Results:
> - cosine(Embd_Q, Embd_D1) = 0.263
> - cosine(Embd_Q, Embd_D2) = 0.286
> - cosine(Embd_Q, Embd_D3) = 0.662
> ```
> where `Embd_Q` is the query embedding, `Embd_D1, Embd_D2, Embd_D3` are doc embeddings generated by our full model, model without key circumstances, and model without key elements, respectively.
>
> We will add this case study with more details to our camera-ready version.
>
> ### 2. Why loss function should be multi-class?
> **A:** In fact, this is one of the main contributions of our work, because we realize the relevance between two legal cases can not be simply specified by 'positive' or 'negative'. Instead, there are different degrees of 'positive' or 'negative'. Our proposed BCL loss can help model recognize the fine-grained relevance, which pushes\pulls doc embeddings to a different distance depending on their relevance to query embeddings.
>
> ### 3. The transitive rule seems to lead to inconsistency?
> **A:** That is a really good question! We acknowledge that this problem does exist. However, according to our experiment, the transitive rule applys to most situations (>92%) where case A,B,C all belong to a single class. Even in the multi-class situation, there are only part of the cases where the transitive rule doesn't work, so the influence on the overall performance is very limited. We will leave it as the future work and discuss its influence in the experiment section.
>
> ### 4. About the writting of sections
> **A:** Sorry for the confusing, we will reorganize and improve Section 4.1&4.2.

---

### Meta-Review · Area_Chair_wq8Y · 2023-09-19

**Recommendation:** 5

**Metareview:**

Reviewers find the paper sound and exciting (3+ for both).  They note the novelty of the task and the clearness of task definition.  A noted concern is the broad applicability of this research methodology to outside of the legal domain, as well as other legal jurisdictions than the area studied.

---

### Decision · Program_Chairs · 2023-10-07

**Decision:**

Accept-Main

**Comment:**

Reviewers find the paper sound and exciting (3+ for both).  They note the novelty of the task and the clearness of task definition.  A noted concern is the broad applicability of this research methodology to outside of the legal domain, as well as other legal jurisdictions than the area studied.